# A Snapshot of Vitamin D Status, Performance, Blood Markers, and Dietary Habits in Runners and Non-Runners

**DOI:** 10.3390/nu16223912

**Published:** 2024-11-15

**Authors:** Francesco Pegreffi, Sabrina Donati Zeppa, Marco Gervasi, Eneko Fernández-Peña, Giosuè Annibalini, Alessia Bartolacci, Eugenio Formiglio, Deborah Agostini, Claudia Barbato, Piero Sestili, Antonino Patti, Vilberto Stocchi, Rosa Grazia Bellomo

**Affiliations:** 1Department of Medicine and Surgery, School of Medicine and Surgery, “Kore” University of Enna, 94100 Enna, Italy; francesco.pegreffi@unikore.it; 2Department of Biomolecular Sciences, University of Urbino Carlo Bo, 61029 Urbino, Italy; sabrina.zeppa@uniurb.it (S.D.Z.); giosue.annibalini@uniurb.it (G.A.); a.bartolacci2@campus.uniurb.it (A.B.); e.formiglio@campus.uniurb.it (E.F.); deborah.agostini@uniurb.it (D.A.); c.barbato2@campus.uniurb.it (C.B.); piero.sestili@uniurb.it (P.S.); rosa.bellomo@uniurb.it (R.G.B.); 3Department of Physical Education and Sport, University of the Basque Country UPV/EHU, 01007 Vitoria-Gasteiz, Spain; eneko.fernandezp@ehu.eus; 4Department of Psychology, Educational Science and Human Movement, University of Palermo, 90144 Palermo, Italy; antonino.patti01@unipa.it; 5Department of Human Sciences for the Promotion of Quality of Life, University San Raffaele, 20132 Roma, Italy; vilberto.stocchi@uniroma5.it

**Keywords:** vitamin D, leucocyte, endurance, runners, healthy people, vo_2_max, maximal isometric force, monocyte

## Abstract

Background: Vitamin D can influence athletic performance and infection risk. This study aimed to investigate vitamin D status, hematochemical factors, anthropometric and performance parameters, and dietary habits in runners (n = 23) and sedentary healthy individuals (non-runners, n = 22) during the autumn season. Methods: Both groups had their serum 25-Hydroxyvitamin D (ng/mL) levels, blood and performance parameters, and dietary habits measured. Results: Serum 25-Hydroxyvitamin D levels were significantly lower in non-runners (runners: males 30.0 ± 5.6, females 31.2 ± 5.2 vs. non-runners: males, 22.8 ± 6.5, females 24.7 ± 6.5 ng/mL, *p* < 0.001). White blood cells, monocyte, and neutrophil levels were higher in non-runners for both males and females. Among the subjects, 23 had optimal vitamin D levels (>29 ng/mL), while 22 had insufficient/deficient levels (<29 ng/mL), with a higher prevalence of insufficiency in non-runners compared to runners (63.6% vs. 34.8%; *p* = 0.053). Maximal isometric force and jump height were equal in both groups, but VO_2_max was higher in runners. Linear regression analysis identified monocyte count as the only predictor of vitamin D levels for both males (y = −24.452 x + 40.520; R^2^ = 0.200; *p* = 0.015) and females (y = −33.409 x + 45.240; R^2^ = 0.368; *p* = 0.003). Conclusions: This study highlights significant differences in vitamin D status between runners and non-runners, with runners exhibiting higher serum 25-Hydroxyvitamin D levels, although this finding is likely due to the increased sun exposure that runners receive. It also provides valuable insights into the vitamin D status of healthy young sedentary individuals and runners, enhancing the understanding of how physical activity influences vitamin D levels.

## 1. Introduction

Vitamin D is a fat-soluble secosteroid hormone primarily obtained by exposure to ultraviolet B rays and secondarily by nutrition. UVB radiation may produce around 80% of this vitamin by converting 7-dehydrocholesterol to previtamin D. Ergocalciferol or cholecalciferol can be obtained by food. These levels might vary based on season, sun exposure duration, and ethnicity, among other things [1]. In addition to effects linked to calcium homeostasis and bone metabolism, vitamin D has been linked to other pleiotropic effects. It is widely believed that the interaction between 1–25 dihydroxy cholecalciferol (1,25(OH)2D3) and a nuclear vitamin D receptor (VDRn), which when liganded forms a heterodimeric complex with the retinoid-X receptor (RXR), mediates the actions of 1,25(OH)2D3. By attaching to target gene promoter sequences known as the vitamin D response element, this complex has the ability to either up- or down-regulate the transcription of many target genes [2]. Alongside the genomic pathway, a non-genomic pathway has also been proposed. These activities, which may include both VDRn and a membrane VDR, show up as the activation of signaling cascades or pathways that cause cells to react right away. They may also have an influence on general physiological processes by influencing epigenetic regulation [3]. Although some of the proposed mechanisms for non-genomic actions have only been described in vitro, and the actual role in vivo has yet to be demonstrated, they could help elucidate the potential of vitamin D as a regulator.

Vitamin D is essential for the immune system, cardiovascular health, and musculoskeletal health [4,5]. Furthermore, endothelial dysfunction, dyslipidemia, type 2 diabetes, and cardiovascular illnesses have all been linked to its lack [6]. Poor diet reduced cutaneous vitamin D synthesis (e.g., decreased sun exposure), altered expression of vitamin D metabolic enzymes, and decreased expression of the VDR in skeletal muscle are associated with an increased risk of insufficiency and abnormal vitamin D activity [7]. The degree to which VDR is expressed varies among tissues, determining the vitamin D’s degree of effect [8].

Vitamin D plays a crucial role in the formation and upkeep of robust bone structure, with the underlying mechanisms governing its effects well elucidated. Specifically, the active form of vitamin D, 1,25(OH)2D3, facilitates the absorption of calcium and phosphate through diverse pathways, thereby enhancing bone mineralization and serving as a protective measure against bone deterioration and fractures. VDR was also found in skeletal muscle tissue [9], indicating that vitamin D may affect muscle in both physiological and pathological conditions [10,11]. Skeletal muscle oxidative stress, which impacts mitochondrial activity and contributes to the development of skeletal muscular atrophy, is linked to vitamin D deficiency (low serum of 25-Hydroxyvitamin D: 25(OH)D), and the activation of VDR may be associated with these detrimental effects. Furthermore, vitamin D deficiency may exacerbate muscular atrophy [12], and it is also associated with a lower level of muscle function, as well as an increased risk of sarcopenia and other illnesses [13].

Conditions involving VDR knockout and vitamin D deficiency appear to imply detrimental effects on the homeostasis of skeletal muscle. In C2C12, until the cells fully mature into myotubes, VDR expression is gradually lowered from a high level at the start of the differentiation process. Intracellular VDR concentration is higher in undifferentiated cells than in differentiated cells, according to a prior study by Kong et al. [14].

Given that several studies conducted on humans have linked inadequate levels of vitamin D to a decrease in muscular function [13], sports science is becoming increasingly interested in the potentiality of vitamin D in maximizing athletic performance [15]. Despite not much research on vitamin D in endurance athletes, a positive correlation between serum 25(OH)D concentration and maximum oxygen consumption (VO_2_max) (r = 0.29, *p* = 0.0001) has been demonstrated. Additionally, a significant interaction (*p* < 0.02) was discovered between self-reported hours of moderate to intense physical activity and 25(OH)D level [16]. The predominant focus of the existing literature on vitamin D and sports performance has centered on exploring the association between its levels and muscular strength and power. Supplementation with vitamin D has been shown to notably enhance muscle strength and power in athletes, with a pronounced impact observed on lower-body muscular strength as opposed to upper-body muscle strength [17]. Conversely, insufficient vitamin D levels have been associated with a lower physical performance in indoor athletes [18,19,20]. However, the impact of vitamin D levels on athletic performance remains unclear across various populations, including outdoor athletes and healthy individuals, due to the significant variability in study outcomes.

One billion people are thought to be deficient in vitamin D globally [21]. A serum 25(OH) D concentration of less than 20, 20 to 29, and more than 29 ng/mL has been proposed to characterize vitamin D deficiency, insufficiency, and sufficiency, respectively [21]. Even while supplementation with vitamin D is often advantageous, there is no agreement on the suggested daily amount of vitamin D, and little information is presently available regarding vitamin D intake among young adult demographic groups, including university students.

Although data on individuals with adequate vitamin D levels is limited, the advantages of supplementation in enhancing energy metabolism, muscle mass, and strength among those deficient in vitamin D are extensively acknowledged [1]. An examination of the resting metabolic rate, strength, and body composition of physically active young people who were sufficient in vitamin D and took a 12-week supplement found no further physiological advantages, with blood total 25(OH)D concentrations reaching supraphysiological levels [22]. In addition, vitamin D is crucial for the control of inflammatory response and immune system. According to recent research, immune cells, including monocytes, macrophages, dendritic cells, and lymphocytes, express VDR and enzymes that activate vitamin D and respond to vitamin D lowering the levels of pro-inflammatory cytokines [23,24].

According to Jones et al., vitamin D status modulates exercise-induced alterations in innate immune defense parameters and metabolomic signatures, including indicators of inflammation and metabolic stress [25]. It has been discovered that low vitamin D status has detrimental impacts on immunological health in athletic populations, which supports the advice that athletes’ circulatory 25(OH)D concentrations should be closely monitored [25].

It may be possible to obtain insight into the effects of various lifestyle choices on vitamin D status and its link to the immune system by comparing the experiences of athletes and non-athletes. This exploratory study aimed to investigate vitamin D levels, blood markers associated with immunological function, nutritional status, cardiorespiratory fit-ness, strength performance, and the correlation between these parameters in runners and non-runner healthy subjects. Our hypothesis is that an endurance sport such as running can help improve vitamin D status in healthy subjects, and that vitamin D levels can influence performance and the immune system.

## 2. Materials and Methods

### 2.1. Participants

Five local running associations were contacted and informed about the research project and inclusion/exclusion criteria. Out of thirty-two responses, seven runners were excluded for not meeting the criteria (four due to age and three due to insufficient weekly training volume). Additionally, two runners were excluded: one due to insufficient training caused by work commitments, and the other because they had the flu in the days leading up to the blood test. non-runners were also contacted through social media and local cultural associations. From the 25 responses received, three individuals were excluded for exceeding the maximum age limit. Ultimately, a cohort comprising forty-five individuals consisting of twenty-three amateur runners (fifteen males, eight females) and twenty-two sedentary healthy subjects (ten males, twelve females) completed this study. All participants were Caucasian and lived and trained at a latitude between 43.6° N and 43.9° N. The inclusion criteria were being healthy at the time of the study and aged between 25 and 45 years for all participants. For athletes, the eligibility criteria required at least three years of continuous training in endurance running prior to the start of the study.; a training frequency of at least three times/week; minimum average mileage outdoors of at least 50 km/week for men and 40 km/week for women. Exclusion criteria included being a smoker; consuming more than three alcoholic drinks per day; acute or chronic disease or treatment with drugs affecting muscle recovery and musculoskeletal performance; use of supplements such as vitamin D, calcium, iron or immune-stimulating complexes containing zinc or echinacea; and, for female subjects, early menopause. All participants provided written informed consent to participate in this study, following a medical health screening. The protocol was approved by the Ethics Committee of the University of Urbino “Carlo Bo”, Italy (54_24gennaio2023_running D+) and was conducted in accordance with the Declaration of Helsinki for research with human volunteers. All data were collected during the 2023 autumn season (third week of October). Data about temperature, average daily solar radiation, and daily sunshine duration were obtained from the Osservatorio Metereologico “Alessandro Serpieri”, Department of Applied and Pure Sciences of the University of Urbino (Italy).

### 2.2. Experimental Design and Procedures

After the recruitment phase, all participants were invited to the laboratories of the research center and followed the same routine: (i) compilation of the Physical activity rating (PA-R) 0–15 scale questionnaire [26]; (ii) anthropometric assessments (weight, height); (iii) warm-up phase consisting of 5 min of walking/running treadmill followed by 5 min of mobilization exercises; (iv) a maximal isometric strength test using a sensor-controlled leg press; (v) after a 5 min rest, a countermovement jump test by using a force platform. Participants underwent serum analysis to measure vitamin D levels, blood count, leukocytes, high-sensitivity C-reactive protein (hs-CRP), calcium, and iron serum levels one week after the physical tests to avoid any acute influence on blood values.

### 2.3. Jump Test

Vertical peak force (N) produced during vertical countermovement jumps (CMJ) performed with both legs and without any arm movement was measured using a force platform (MuscleLabTM system, type PFMA 3010e, Ergotest Innovation, AS, Stathelle, Norway) at a sampling rate of 100 Hz. Each participant started from a stationary erect position with knees fully extended. The participants then squatted down to about 90- of knee flexion before starting the upward motion. Subjects were instructed to keep their hands on their hips to prevent the influence of arm movements. The position of the feet was standardized during all tests at shoulder width. Participants were carefully observed before and during the jumps to ensure that the proper placement and jumping technique were used, and only correct trials were accepted. Subjects performed three to five warm-up jumps and then started a total of three maximal jumps. The highest jump determined by the impulse-momentum method was collected [27].

### 2.4. Isometric Leg Press Test

Peak force (in N) developed during closed-chain maximal isometric contractions was measured using a load cell (AIP, Varese, Italy) connected to an A/D converter (MuscleLabTM system) mounted on a horizontal leg press (Technogym, S.p.A, Cesena, Italy) at a sampling rate of 100 Hz. The load cell was positioned in series with the sliding axis of the leg press so that the direct line of force was registered. Before each trial, the two chains fixing the load cell to the leg press were tensed to obtain a rigid system. Then, the load cell was reset to zero to negate the force produced on it by the two chains. The dynamometer was routinely calibrated using ISO-certified weights. The backrest of the leg press, on which the subjects were lying, was inclined 30° from the horizontal plane. The knee angle was set at about 100- and was controlled using an electronic goniometer (MuscleLabTM system). Before the maximal isometric contractions, subjects performed two to three submaximal isometric contractions as specific warm-up and practice. The participants then performed three maximal isometric contractions, with 2–3 min of recovery in between. Subjects were asked to exert force as hard and fast as possible for 5 s. During isometric leg-press tests, participants were verbally encouraged. The maximal of the peak forces measured during the three maximal trials was used as maximal isometric force [28].

### 2.5. Maximal Oxygen Consumption Estimation

Maximal oxygen consumption was estimated for all participants following the equation proposed by Jamnick et al. (2016) [26]:Estimated VO_2_ max (mL/kg/min) = 56.363 + (1.921 × PA-R) – (0.381 × Age) – (0.754 × BMI) + (10.987 × Gender)
where PA-R is the Physical activity rating (PA-R) questionnaire assessed on a scale from 0 to 15; Age is participants’ age in years; BMI is the Body Mass Index in kg/m^2^; and Gender is 0 for females and 1 for males.

### 2.6. Dietary and Training Monitoring

The participants’ diet and training was monitored for the two weeks prior to serum analysis. They were asked to keep a food and training diary, and an experienced nutrition researcher performed a daily 24-hour recall to remind them to fill in the diary and ensure it was done correctly. The participants were instructed on how to properly complete the food diaries, including methods for measuring their food. Subjects were encouraged to include all food and beverages consumed over the course of the fourteen-day period in their food diary as detailed and accurate as possible. In fact, the food diary must contain date, time and weight (grams) of all the meals consumed. The missing data from the questionnaire and clarifications on food consumed/portions were followed up via email or phone call.

Food diary information was collected and then processed using WinFood nutritional analysis software version Pro 3.37.3 (Medimatica S.u.r.l., Teramo, Italy), and macro- and micronutrients, particularly average calcium and vitamin D intakes, were included in the analyses. The daily intake of macronutrients (proteins, lipids, and glucides) was reported in grams (g), percentage (%), and grams per kilogram of body weight (g/kg/day). Regarding micronutrients, they are expressed in milligrams (mg) and micrograms (μg).

Training diary was completed with information about training type (continuous running or interval training), volume (in km), and rate of perceived exertion (RPE) for each session (CR-10).

### 2.7. Blood Analysis

One week after the anthropometric and performance tests, all participants were invited to a blood testing center (Biolab s.r.l., Pesaro, Italy) to undergo blood sampling. All participants were asked to present themselves by 8.30 a.m. with an empty stomach for at least 8 h. The laboratory analyses were performed blind, i.e., the operator was unaware of the group to which the subject belonged (runners and non-runners). Patients were made to sit in an inclined chair, and blood samples were taken using a butterfly needle from the antecubital vein in the arm. Blood levels of 25-(OH)D3 were quantified by means of a chemiluminescence immunoassay (Beckman Coulter; A98856, https://www.beckmancoulter.com/products/part/A98856, accessed on 1 February 2023) on serum samples. The assay has the following intra-laboratory accuracy: SD ≤ 1.5 ng/mL at concentrations ≤ 15.0 ng/mL; CV ≤ 10.0% at concentrations > 15.0 ng/mL (37.5 nmol/L). The other hematochemical variables were analyzed using a Beckman Coulter automated analyzer. The hematochemical parameters assessed were as follows: White blood cell count, Lymphocytes, Monocytes, Neutrophil granulocytes, Eosinophil granulocytes, Basophil granulocytes, Lymphocytes (absolute number), Monocytes (absolute number), Neutrophils (absolute number), Eosinophils (absolute number), Basophils (absolute number), Calcium (Ca) serum electrolyte, and hs-CRP.

### 2.8. Statistical Analysis

All statistical analyses were performed using JASP (version 0.18.2.0, Department of Psychological Methods, University of Amsterdam, Amsterdam, The Netherlands, https://jasp-stats.org/). The data were assessed for normality using the Shapiro–Wilk test and for homogeneity of covariance matrices using Box’s M test. A two-way multivariate analysis of variance (MANOVA) was conducted to evaluate the differences in the dependent variables according to the group (runners vs. non-runners), gender (males vs. females), and the interaction between gender and group. Chi-squared tests were used to assess the percentage distribution among male/female, optimal/insufficient, and runners/non-runners groups [29,30,31]. Correlations were examined using Pearson or Spearman tests, depending on the normality of the data for both males and females, controlling for Body Mass Index (BMI) and age. An independent samples *t*-test or non-parametric equivalents (Mann–Whitney U Test) were used to assess differences in nutrient intake based on running activity (runners vs. non-runners). All data are presented as mean ± SD. Effect sizes were measured using Cohen’s d [32], with the following interpretations: small (d = 0.2), medium (d = 0.5), and large (d = 0.8). Finally, multiple linear regression was employed to estimate the relationship between the most correlated independent variables and vitamin D as the dependent variable for males and females, with the goodness of fit of the linear regression models expressed as adjusted R². Statistical significance was set at *p* < 0.05.

## 3. Results

### 3.1. Subject Characteristics and Performance Differences Between Runners and Non-Runners

The non-runners and runners had the same age (runners: males 35.7 ± 7.2, females 30.9 ± 9.2; non-runners: males 32.2 ± 7.1, females 28.5 ± 4.7, *p* = 0.055) and both groups had a healthy BMI, even if the athletes exhibited a significantly lower BMI when compared to their control counterparts (see Table 1). The male runners exercised on average 6.1 ± 0.7 sessions per week, with 49.7 ± 12.6 km of continuous running (RPE = 3.1 ± 1.0) and 11.5 ± 2.1 km of interval running (RPE = 7.3 ± 1.0). The female runners exercised on average 4.9 ± 0.9 sessions per week, with 35.1 ± 11.8 km of continuous running (RPE = 3.2 ± 1.2) and 7.8 ± 0.8 km of interval running (RPE = 6.8 ± 0.7). These outdoors training sessions were performed during daylight, and the weather was on average pretty sunny, with 20.0 ± 1.7 °C of average temperature, 15745.8 ± 1033.7 W/m^2^ of average daily solar radiation and 9.6 ± 0.8 h of daily sunshine duration.

There was a large difference between the VO_2_max values in the two groups and between males and females (*p* < 0.001), with a VO_2_max of 62.5 ± 3.8, 51.5 ± 3.5, 47.3 ± 7.4 and 40.6 ± 6.6 mL/min/kg in male runners, female runners, male non-runners, and female non-runners, respectively (Table 1 and Figure 1). Maximal isometric force (MIF) was higher in males compared to females, with no significant differences observed between the groups or their interactions. Similar results were found for jump height (see Table 1).

### 3.2. Hematological Differences Between Runners and Non-Runners

Blood serum 25-Hydroxyvitamin D levels were significantly lower in non-runners, with no differences observed based on gender or interaction (runners: males 30.0 ± 5.6, females 31.2 ± 5.2 vs. non-runners: males 22.8 ± 6.5, females 24.7 ± 6.5 ng/mL, *p* < 0.001). No significant differences were found in blood calcium levels between groups, between males and females, or in their interaction (see Table 1 and Figure 1). White blood cell, monocyte, and neutrophil levels were found to be significantly higher in non-runners than in runners (see Table 1 and Figure 1). The hs-CRP levels were found to be not significantly different between runners and non-runners, between genders, or in their interaction.

### 3.3. Nutritional Differences Between Runners and Non-Runners

The caloric intake of runners was higher than non-runners (1975.0 ± 427.8 Kcal vs. 1679.4 ± 377.0 Kcal; *p* = 0.023, ES 0.731). In particular, protein intake (87.2 ± 18.4 g vs. 69.9 ± 18.7 g) (*p* = 0.004, ES 0.933), available carbohydrates (239.9 ± 63.6 g vs. 196.0 ± 46.5 g; *p* = 0.015), and total fiber (18.4 ± 6.4 g vs. 13.7 ± 5.4 g; *p* = 0.006) were higher in runners than in non-runners (Table 2). The daily relative protein (1.6 ± 0.3 g/kg/day vs. 1.3 ± 0.4 g/kg/day; *p* = 0.003) and carbohydrates (4.5 ± 1.1 g/kg/day vs. 3.6 ± 0.9 g/kg/day; *p* = 0.005) intake was also higher in runners than in non-runners, but not lipids daily relative intake (Table 2). Regarding micronutrients, phosphorus and iron intake was higher in runners (1204.8 ± 301.8 mg vs. 975.9 ± 278.7 mg, *p* = 0.015; 11.9 ± 3.3 mg vs. 9.3 ± 3.1mg, *p* = 0.012) compared to non-runners. Another significant result concerns the intake of thiamine, which, also in this case, is higher in runners (1.1 ± 0.3 mg) than in non-runners (0.8 ± 0.2 mg) (*p* ≤ 0.001). However, a general reduction in vitamins intake is observed in runners compared to non-runners (vitamin A; beta-carotene, vitamin B5, B8, and K *p* < 0.05). Regarding the remaining variables (water, proteins in %, lipids in g and in %, available glucides in %, starch, oligosaccharides, saturated fatty acids, unsaturated fatty acids, soluble fibers, insoluble fibers, animal and vegetable proteins, calcium, sodium, magnesium, potassium, folic acid, niacin, riboflavin, alpha-tocopherol, vitamin C, vitamin D, vitamin E, vitamin B6, vitamin B12, total polyphenols, total-Orac) there are no differences between runners and non-runners (Table 2 and Figure 1).

### 3.4. Vitamin D Levels Between Optimal and Insufficient/Deficient Groups

The characteristics of subjects with optimal and insufficient/deficient vitamin D levels were compared. Specifically, it was found that 23 subjects had optimal 25(OH)D levels (>29 ng/mL), while 22 had insufficient/deficient 25(OH)D levels (<29 ng/mL). Interestingly, the frequency of insufficient subjects was higher in non-runners than in runners (63.6% vs. 34.8%; *p* = 0.053). When comparing runners and non-runners, only eight of the non-runners (36.4%), but fifteen of the runners (65.2%), had optimal levels. In contrast, fourteen non-runners (63.6%) and only eight runners (34.8%) had inadequate vitamin D levels (Table 3). 

### 3.5. Pearson’s Correlation Results

Following Pearson correlation testing, 25(OH)D showed a weak but significant positive correlation with VO2max for males only (males: r = 0.406, *p* = 0.049; females: r = 0.379, *p* = 0.110). There was no significant correlation with BMI (males: r = −0.344, *p* = 0.100; females: r = −0.329, *p* = 0.169). For females, a significant negative correlation was found with white blood cells (males: r = −0.282, *p* = 0.182; females: r = −0.589, *p* = 0.008) and neutrophils (males: r = −0.229, *p* = 0.282; females: r = −0.619, *p* = 0.005). Monocytes showed a significant negative correlation for both males (r = −0.462, *p* = 0.023) and females (r = −0.647, *p* = 0.003). Additionally, hs-CRP was negatively correlated only in males (r = −0.403, *p* = 0.050; females: r = −0.258, *p* = 0.286). Conversely, weak but non-significant negative correlations were observed between VO_2_max and white blood cells in males (males: r = −0.424, *p* = 0.050; females: r = −0.249, *p* = 0.305). Significant inverse correlations with VO_2_max were found for neutrophils and monocytes only in females (neutrophils: males: r = −0.329, *p* = 0.116; females: r = −0.470, *p* = 0.042; monocytes: males: r = −0.388, *p* = 0.066; females: r = −0.570, *p* = 0.011). Moreover, monocytes correlated inversely with hs-PCR in males (r = 442, *p* = 0.050). Finally, MIF showed a positive correlation with BMI only in females (males: r = −0.343, *p* = 0.100; females: r = −0.571, *p* = 0.011).

### 3.6. Multiple Linear Regression Results

A stepwise multiple linear regression analysis was performed for both males and females to calculate the estimated relationship between 25(OH)D levels and the measured anthropometric, performance, and blood parameters. The covariates considered but not included were lymphocytes, neutrophils, eosinophils, maximal isometric force, VO_2_max, BMI, and age. This model shows that monocyte count is the only identified predictor of 25(OH)D levels both for males (y = −24.452 x + 40.520; R^2^ = 0.200; *p* = 0.015) and females (y = −33.409 x + 45.240; R^2^ = 0.368; *p* = 0.003). The other analyzed parameters were not significantly associated with 25(OH)D levels. The goodness-of-fit index R^2^ values are 0.200 for males and 0.368 for females, indicating that approximately 20% of the variation in 25(OH)D levels for males and 37% for females can be explained by monocyte levels (Figure 2).

## 4. Discussion

A growing body of evidence suggests that vitamin D could play a valuable role in athletic performance. Vitamin D interacts with extra-skeletal tissues, modulating injury, inflammation, and infection risk. Poor vitamin D status is well-known to be widespread throughout Europe, and, notably, it has also been observed in the professional sports world [4]. This exploratory study aimed to investigate the vitamin D status in Italian runners (n= 23) and sedentary healthy individuals (non-runners, n = 22) in the autumn period. An insufficient vitamin D status has been observed in 34.8% of runners and 63.6% of non-runners.

The findings revealed that blood calcium levels did not differ between runners and non-runners, gender, or their interaction. In contrast, serum vitamin D levels were considerably lower in the non-runners group compared to runners. No significant differences were found between genders or in their interaction. Similar findings were reported by Çetin Daglı et al. (2023), who analyzed vitamin D levels in sedentary people and athletes who competed in both indoor and outdoor sports and found that outdoor activities had a greater beneficial impact on blood lipid profiles and vitamin D3-25-OH levels than indoor sports [33]. A weak positive but significant correlation was found between vitamin D and VO_2_max for males only (r = 0.406, *p* = 0.049), which is in agreement with the results obtained by Ardestani et al [16]. However, this correlation is likely attributed to the increased sun exposure runners receive, as they engage in three to six outdoor running sessions per week. Notably, the vitamin D intake was similar in both groups, reinforcing the idea that outdoor activities significantly contribute to higher vitamin D levels. Furthermore, the maximal isometric force test and jump height showed no difference between runners and non-runners, nor any interaction with gender or correlation with vitamin D levels. This suggests that the serum vitamin D levels reported in this study are unlikely to have an ergogenic effect on either endurance or strength performance, as has been reported by other authors for different supplements [34].

Reduced serum levels of vitamin D are associated with an increased susceptibility to a range of conditions, including cancer, cardiovascular disorders, and neurological ailments. These diseases are typified by the presence of underlying chronic inflammation, which potentially contributes to the onset and progression of the respective pathologies. In this regard, Bellia et al. (2013) [35] identified an inverse association between vitamin D levels and serum inflammatory markers, including C-reactive protein, interleukin-6, and Tumor Necrosis Factor α [35,36]. In accordance with this evidence, we found that hs-CRP is inversely related to vitamin D level, but only in males. However, there is no significant difference in hs-CRP levels between runners and non-runners, nor between genders or their interaction, likely due to the large variability in hs-CRP values. Therefore, further research is needed to assess whether outdoor exercise can prevent the development of inflammatory diseases.

Vitamin D regulates the inflammation response and immune system through VDR and enzymes that activate vitamin D and respond to vitamin D, lowering the levels of pro-inflammatory cytokines [23,24,37,38]. A powerful regulator of the transcriptome of hundreds of genes and the epigenome of genomic sites is the complex 1,25(OH)2D3-VDR. Monocytes and other innate and adaptive immune cells are a key target tissue [39]. Neme et al. found 702 genes that are strongly impacted by vitamin D and are involved in protein translation, monocyte differentiation, and cellular growth regulation after analyzing the in vivo transcriptome of human white blood cells in response to the nutrient [40]. According to our findings, leukocyte counts in the two groups revealed a notably different distribution of neutrophils, monocytes, and white blood cells, with non-runners having larger amounts. Specifically, monocytes had a greater significant effect (*p* value < 0.001) showing an inverse correlation with the vitamin D status in both males and females (see Table 1).

This study underscores the clinical and physiological significance of the differences observed, particularly in white blood cell counts, between runners and non-runners, where higher levels in non-runners align with prior findings on endurance athletes [41,42]. While this study presents a cross-sectional view of vitamin D and immune markers across the two groups, the focus on statistically significant results provides a basis for further exploration. Notably, the elevated white blood cells in non-runners might indicate physiological adaptations to lower physical activity levels, influenced by immune responses. Although direct clinical guidelines are not established, these findings suggest that physical activity could shape immune profiles. Longitudinal studies with larger sample sizes are necessary to clarify these implications for athletic health and performance.

Monocytes, important cells of the innate immune system that can differentiate into macrophages and dendritic cells, have been found to be moderately inversely related to vitamin D level and VO_2_max and slightly positive with hs-CRP. Our results demonstrated an inverse relationship between leukocytes, neutrophils, monocytes, and vitamin D. However, multiple regression analysis reveals that monocytes are the only predictor of variance (20% for males and 37% for females) in vitamin D levels.

Similarly to our results, Tang et al. demonstrated that vitamin D is inversely associated with monocyte to HDL-C ratio [43] and hypothesized that inverse association between vitamin D and MHR may be elucidated by several underlying mechanisms. First, by down-regulating adhesion molecules such as PSGL-1, β(1)-integrin, and β(2)-integrin, vitamin D can lower monocyte activation [44]. Furthermore, in a dose- and time-dependent manner, vitamin D inhibited the expression of mRNA and receptor proteins in human monocytes [45]. Our manuscript further highlights that, while our findings suggest a possible link between elevated vitamin D and reduced monocyte levels, this association should be interpreted cautiously. We aim to underscore the need for longitudinal research to determine causality and better understand the immune-modulating effects of vitamin D.

Of the 45 participants, 23 had optimal vitamin D levels, and 22 had insufficient/deficient levels (>29 ng/mL and <29 ng/mL, respectively) [31]. There were no variations in the frequency of gender-specific vitamin D levels (Table 3). Rather, there was an opposite trend between runners and non-runners: non-runners had a higher percentage of insufficient/deficient levels of vitamin D (63.6% vs. 36.8%), while the latter had a higher percentage of optimal vitamin D (65.2% vs. 34.8%).

A healthy lifestyle is based on a correct diet and regular exercise, in particular outdoor sports. Several studies suggest that regular physical activity can positively affect eating behavior and food choices in a healthy direction, helping in correcting inappropriate long-term eating habits [46]. Donati Zeppa et al. [47] reported that nine weeks of HIIT promoted a spontaneous modulation of food choices and regulation of dietary intake in young normal-weight subjects. The monitoring of the nutritional habits of the enrolled subjects has revealed that runners ingest more calories than non-runners, most likely because physical activity involves a significant energy demand [48]. A higher intake of proteins, available glucides (in both absolute values and daily relative values) and total fiber have been observed in our runners. However, the daily fiber intake fell short of the recommended 30 grams per day. [49].

The protein intake is in line with contemporary nutritional recommendations for middle-distance runners that train 40–180 km/week during the autumn/winter session (1.5–1.7 g protein/kg/day), while carbohydrate intake is slightly lower (4.5 ± 1.1 g/kg/day vs. recommended 7–10 g/kg/day) [50]. This difference can be explained by considering that our participants were not elite athletes but amateur runners who often have self-planned eating habits. This result highlights that amateur runners often do not have an optimized diet regarding carbohydrate or fiber intake and would probably benefit from professional counseling. However, the notable difference in carbohydrate and protein intake between our runners and non-runners underscores the greater energy and nutritional demands required by the former. These demands are essential for sustaining the volume and intensity of training that ultimately leads to their higher VO_2_max.

The European Food Safety Authority (EFSA) [51] reported that vitamin D is primarily synthesized by our body, accounting for approximately 80% of the 600 IU daily requirement. The remaining 20% is obtained through dietary intake. The proportions of vitamin D derived from dietary sources are consistent between runners and non-runners. Given the comparable dietary intakes of vitamin D in both groups, the difference in blood vitamin D levels between runners and non-runners is attributed to the endogenous vitamin D produced from 7-dehydrocholesterol through sun exposure [51].

A higher water and electrolyte intake has also been observed in runners with respect to the control; a correct hydration and electrolyte balance is important for sustaining cognitive and physical performance through the regulation of muscle contraction, fluid balance, and nerve impulses conduction (Table 2). The idea that the serum vitamin D difference between runners and non-runners is caused by endogenous synthesis is supported by the fact that no changes in vitamin D consumption have been found between the two groups.

This study has some limitations and strengths. Direct measurement of solar radiation exposure for participants was not conducted, which prevents accurate quantification and individualization of the effect of outdoor physical activity on vitamin D status. However, since the runners exercised in good weather during daylight hours, it is likely that the observed differences were due to outdoor exercise. The sample size was relatively small. However, despite these constraints, this study was adequately powered (0.81) to detect significant differences in key outcomes, including a significant difference in vitamin D levels between the two groups. Despite these limitations, this study provides a unique comparison of vitamin D status, immune markers, and performance metrics between runners and sedentary individuals during the autumn season, offering insight into seasonal effects on these parameters.

This study provides valuable insights into the vitamin D status of healthy young sedentary individuals and runners, enhancing the understanding of how physical activity influences vitamin D levels. All participants generally found inadequate vitamin D levels, but runners had higher vitamin D levels than reported in the literature for athletes. Furthermore, most subjects with an insufficient/deficient vitamin D level were not runners.

A novel finding of this work is the identification of monocyte count as a significant predictor of vitamin D levels in both male and female participants, suggesting a potential biomarker for understanding vitamin D’s role in immune modulation across different activity levels. In conclusion, the data indicate that both runners and non-runners face a potential risk of vitamin D insufficiency or deficiency during the autumn season. This underscores the need for additional research to explore the complex interplay between low vitamin D levels, immune function, inflammatory responses, and physical performance. Although direct clinical guidelines are not yet established, our findings suggest that physical activity may significantly influence immune profiles. To fully understand these implications for athletic health and performance, future longitudinal studies with larger sample sizes will be essential, providing deeper insight into how vitamin D status and activity levels collectively shape immune function and overall health.

## Figures and Tables

**Figure 1 nutrients-16-03912-f001:**
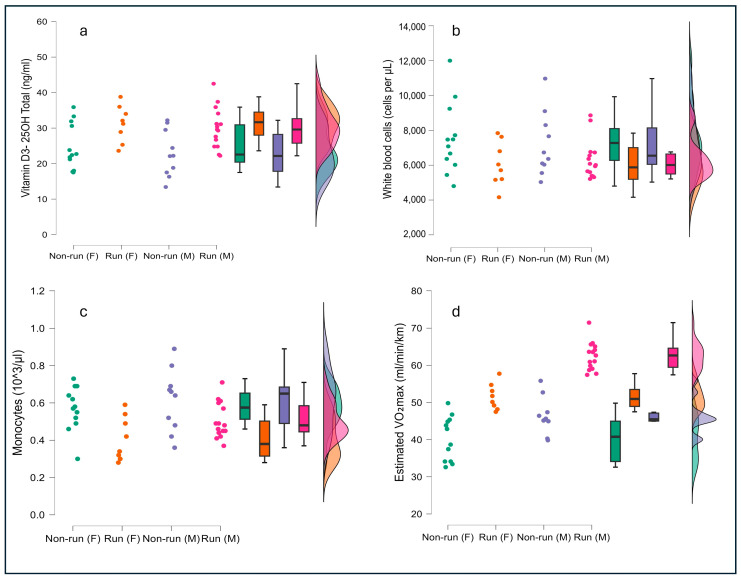
Raincloud plots of (**a**) total vitamin D3-25(OH)D, (**b**) count of white blood cells, (**c**) monocytes, and (**d**) estimated VO_2_max for female non-runners (Non-run (F)), female runners (Run (F)), male non-runners (Non-run (M)), and male runners (Run (M)) groups. The boxplots represent the median and the interquartile range, while the one-sided violin plots represent the smoothed distribution curve.

**Figure 2 nutrients-16-03912-f002:**
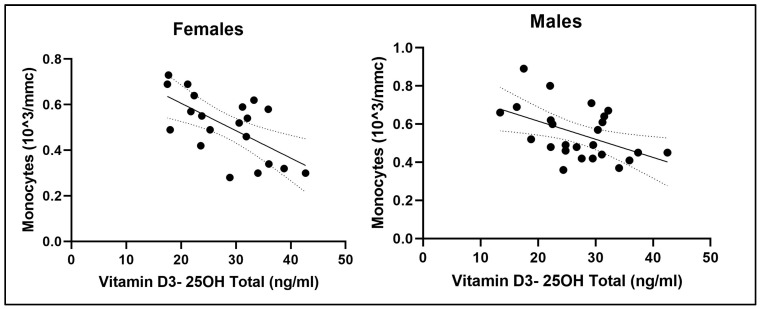
Correlation plot of monocytes on vitamin D3-25(OH) total for females and males.

**Table 1 nutrients-16-03912-t001:** Demographics, performance, and blood parameters: analysis by running activity (runners vs. non-runners), gender, and their interaction.

	Runners (23)		Non-Runners (22)		*p* Runners vs. Non-Runners	*p* Gender	*p* Interaction
Participants	Males = 15	Females = 8	Males = 10	Females = 12			
Age (y)	35.7 ± 7.2	30.9 ± 9.2	32.2 ± 7.1	28.5 ± 4.7	0.177	0.055	0.792
Height (m)	1.7 ± 0.04	1.6 ± 0.05	1.7 ± 0.06	1.6 ± 0.05	0.628	<0.001 ^αα^	0.153
Weight (kg)	65.6 ± 5.6	57.1 ± 7.6	73.9 ± 7.0	56.6 ± 5.2	0.048 ^α^	<0.001 ^αα^	0.025 ^α^
BMI (kg/m^2^)	21.3 ± 1.6	20.7 ± 1.9	23.7 ± 2.1	21.3 ± 1.9	0.016 ^α^	0.013 ^α^	0.149
Est. VO_2_max (mL/min/kg)	62.5 ± 3.8	51.5 ± 3.5	47.3 ± 7.4	40.6 ± 6.6	<0.001 ^αα^	<0.001 ^αα^	0.221
MIF (N)	1692.9 ± 345.7	1371.6 ± 395.6	1732.3 ± 305.8	1280.3 ± 403.2	0.817	0.001	0.560
CMJ (m)	0.31 ± 0.04	0.23 ± 0.03	0.33 ± 0.04	0.23 ± 0.04	0.355	<0.001 ^αα^	0.769
25(OH)D (ng/mL)	30.0 ± 5.6	31.2 ± 5.2	22.8 ± 6.5	24.7 ± 6.5	<0.001 ^αα^	0.395	0.848
Calcium (mg/dL)	9.9 ± 0.3	9.7 ± 0.3	10.1 ± 0.3	9.9 ± 0.3	0.532	0.080	0.532
WBC (cells per µL)	6292.7 ± 1108.5	6067.5 ± 1279.7	7183.0 ± 1831.2	7517.5 ± 2025.2	0.021 ^α^	0.911	0.569
Lymphocytes (10^3^/µL)	2.3 ± 0.4	2.5 ± 0.9	2.6 ± 0.6	2.6 ± 1.0	0.306	0.853	0.625
Monocytes (10^3^/µL)	0.5 ± 0.1	0.4 ± 0.1	0.6 ± 0.2	0.6 ± 0.1	<0.001 ^αα^	0.078	0.501
Neutrophils (10^3^/µL)	3.2 ± 0.9	2.9 ± 0.6	3.7 ± 1.3	4.2 ± 1.5	0.019 ^α^	0.871	0.246
Eosinophils (10^3^/µL)	0.18 ± 0.1	0.23 ± 0.2	0.17 ± 0.1	0.15 ± 0.1	0.172	0.691	0.223
Basophils (10^3^/µL)	0.04 ± 0.01	0.06 ± 0.02	0.05 ± 0.02	0.05 ± 0.02	0.694	0.103	0.227
hs-CRP (mg/L)	1.1 ± 0.8	0.4 ± 0.2	4.148 ± 9.1	2.9 ± 5.6	0.094	0.574	0.862

Values are presented as mean and ± SD. BMI: Body Mass Index; Est. VO_2_max: estimated maximal oxygen consumption; MIF: maximal isometric force; CMJ: countermovement jump; 25(OH)D: serum of 25-Hydroxyvitamin D; WBC: white blood cells; hs-CRP: high sensitivity C-Reactive Protein. *p*-value for gender indicates significant differences in the dependent variables between males and females, regardless of being runners or non-runners; *p*-value for runners vs. non-runners indicates significant differences in the dependent variables between runners and non-runners, regardless of gender; *p*-value for gender interaction runners vs. non-runners: tests if the effect of gender on the dependent variable varies between runners and non-runners. A significant interaction means that the difference in the dependent variable between runners and non-runners is different for males and females. ^α^ = *p* < 0.05, ^αα^ = *p* < 0.001.

**Table 2 nutrients-16-03912-t002:** Participants’ nutrients intake according to running activities (Runners vs. Non-runners).

	Runners (n = 23)	Non-Runners (n = 22)	Test Used	*p* Value	Cohen’s d
Water (g)	744.9 ± 162.3	621.6 ± 254.4	T-Test	0.066	0.584
Calories (kcal)	1975.0 ± 427.8	1679.4 ± 377.0	T-Test	**0.023** ^α^	0.731
Protein (g)	87.2 ± 18.4	69.9 ± 18.7	T-Test	**0.004** ^α^	0.933
Protein (%)	17.9 ± 3.2	16.6 ± 2.2	T-Test	0.129	0.479
Protein (g/kg/day)	1.6 ± 0.3	1.3 ± 0.4	T-Test	**0.003** ^α^	0.943
Lipids (g)	75.6 ± 17.7	67.6 ± 16.2	T-Test	0.136	0.470
Lipids (%)	34.6 ± 4.2	36.4 ± 4.1	T-Test	0.168	0.434
Lipids (g/kg/day)	1.4 ± 0.3	1.2 ± 0.3	T-Test	0.098	0.505
Available glucides (g)	239.9 ± 63.6	196.0 ± 46.5	T-Test	**0.015** ^α^	0.782
Available glucides (%)	48.3 ± 5.0	46.8 ± 5.4	M–W	0.428	0.271
Available glucides (g/kg/day)	4.5 ± 1.1	3.6 ± 0.9	T-Test	**0.005** ^α^	0.895
Starch (g)	142.6 ± 47.3	121.7 ± 35.2	T-Test	0.116	0.497
Oligosaccharides (g)	64.6 ± 23.8	48.5 ± 23.2	T-Test	0.032	0.685
Saturated fatty acids (g)	16.2 ± 4.5	17.3 ± 5.5	T-Test	0.516	0.203
Unsaturated fatty acids (g)	37.9 ± 9.8	38.7 ± 7.7	T-Test	0.765	0.093
Total fiber (g)	18.4 ± 6.4	13.7 ± 5.4	M–W	**0.006** ^α^	0.787
Soluble fiber (g)	2.8 ± 1.2	2.6 ± 1.0	M–W	0.609	0.138
Insoluble fiber (g)	8.0 ± 3.7	6.4 ± 2.8	M–W	0.148	0.468
Calcium (mg)	1012.0 ± 560.0	770.3 ± 373.7	M–W	0.129	0.503
Sodium (mg)	1832.0 ± 610.1	1737.1 ± 466.8	M–W	0.832	0.173
Magnesium (mg)	231.2 ± 117.9	201.4 ± 97.6	T-Test	0.380	0.274
Potassium (mg)	2374.0 ± 574.9	2134.2 ± 686.2	T-Test	0.225	0.381
Phosphorus (mg)	1204.8 ± 301.8	975.9 ± 278.7	T-Test	**0.015** ^α^	0.787
Iron (mg)	11.9 ± 3.3	9.3 ± 3.1	T-Test	**0.012** ^α^	0.809
Folic acid (μg)	160.3 ± 67.1	174.7 ± 64.2	M–W	0.417	0.219
Niacin (mg)	17.1 ± 4.9	14.2 ± 4.7	M–W	0.050	0.613
Riboflavin (mg)	1.7 ± 0.7	1.7 ± 1.3	M–W	0.375	0.006
Thiamin (mg)	1.1 ± 0.3	0.8 ± 0.2	T-Test	**<0.001** ^αα^	1.129
Vitamin A (μg)	455.8 ± 192.5	649.8 ± 61.1	T-Test	**0.034** ^α^	0.680
Beta-carotene (mg)	372.6 ± 385.3	1000.7 ± 862.4	M–W	**0.005** ^α^	0.957
Alpha-tocopherol (mg)	2.8 ± 0.5	2.8 ± 0.6	M–W	0.990	0.074
Vitamin C (mg)	71.1 ± 35.4	63.6 ± 34.4	T-Test	0.489	0.216
Vitamin D (μg)	2.9 ± 1.6	2.6 ± 1.6	M–W	0.627	0.142
Vitamin E (mg)	8.7 ± 3.0	8.4 ± 2.2	M–W	0.852	0.110
Vitamin B5 (mg)	1.0 ± 0.7	1.5 ± 0.7	M–W	**0.004** ^α^	0.785
Vitamin B6 (mg)	1.1 ± 0.4	1.3 ± 0.5	M–W	0.252	0.482
Vitamin B8—Biotin (μg)	3.9 ± 4.1	7.2 ± 4.1	M–W	**0.001** ^α^	0.792
Vitamin B12 (μg)	1.1 ± 0.9	1.1 ± 0.8	M–W	0.931	0.010
Vitamin K (μg)	28.4 ± 39.7	60.4 ± 45.0	M–W	**0.002** ^α^	0.757

Values mean ± SD. Values are based on the average dietary intake during the last 15 days prior to blood analysis and include: water intake, macronutrients in grams and as a percentage of total calories, micronutrients, and trace elements. T-Test: independent *t*-tests; M–W: non-parametric Mann–Whitney U Test. Significant differences are highlighted in bold. *p* value: ^α^ = *p* < 0.05, ^αα^ = *p* < 0.001.

**Table 3 nutrients-16-03912-t003:** Contingence table according to vitamin D status.

	Optimal (>29 ng/mL)	Insufficient/Deficiency (<29 ng/mL)
Combined (n)	23	22
Male (n) (%)	12 (48%)	13 (52%)
Female (n)	11 (55%)	9 (45%)
Runners (n) (%)	15 (65.2%)	8 (34.8%)
Non-runners (n) (%)	8 (36.4%)	14 (63.6%)

Runners vs. Non-runners frequency was tested by Chi-squared tests.

## Data Availability

The raw data supporting the conclusions of this article will be made available by the authors, without undue reservation.

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
