# Peer review of "A Snapshot of Vitamin D Status, Performance, Blood Markers, and Dietary Habits in Runners and Non-Runners"

_nutrients, 2024, doi:10.3390/nu16223912_

Round 1
Reviewer 1 Report
Comments and Suggestions for Authors
The article "A Snapshot of Vitamin D Status, Performance, Blood Markers, and Dietary Habits in Runners and Non-Runners" is marred by several methodological issues and gaps in scientific rigor, which give rise to concerns about its suitability for publication.
1. A significant deficiency is the lack of clarity and consistency in the methodology employed to assess VO2max. Although the article purports to measure VO2max for runners, the protocol employed for non-runners is a submaximal CustomTm test, which yields an estimate of VO2max rather than a definitive measurement. This introduces a significant methodological inconsistency. Estimating VO2max rather than measuring it directly is an inadequate approach for drawing robust conclusions about differences in cardiorespiratory fitness between groups, particularly when VO2max is a central variable in the study.
It is imperative that the text clearly distinguish between measured and estimated values, which is not effectively done. The use of an estimation method calls the validity of the comparison between runners and non-runners into question, as it introduces potential bias that is not adequately addressed.
2. The authors attempt to establish a link between vitamin D levels and performance outcomes, such as VO2max, without providing a scientifically robust mechanism. The correlation between vitamin D and VO2max (r = 0.318) is weak, and the discussion fails to explore potential confounding factors that could explain this observed association, such as sunlight exposure, fitness level, or other nutritional factors.
The assertion that elevated vitamin D levels enhance performance is unsubstantiated and not corroborated by the data presented. The study lacks a clear mechanistic explanation for how vitamin D could directly influence oxygen consumption or endurance performance, beyond what can be attributed to general health status or immune function.
3. The training type was not taken into account.
A significant deficiency is the absence of comprehensive documentation regarding the specific characteristics of the training regimen, including its type, intensity, and duration, for the participants. Although the study differentiates between runners and non-runners, it fails to acknowledge the discrepancies in training regimens, such as endurance training, high-intensity interval training (HIIT), or resistance training, which could all impact VO2max, performance outcomes, and vitamin D levels.
It is essential to report and control for the type and frequency of training in order to gain a comprehensive understanding of performance metrics such as VO2max. In the absence of this information, the study's conclusions regarding the impact of vitamin D on performance are open to question.
4. The study does not adequately control for or report participants' exposure to sunlight, which is the primary source of vitamin D. Given that runners generally train outdoors, their vitamin D levels may be influenced more by sunlight exposure than by diet or other factors. This is an important confounding factor that is not sufficiently addressed in the study.
The authors' failure to account for this variable raises questions about the assertion that vitamin D levels are associated with better performance in runners. The elevated levels of vitamin D observed in runners may merely reflect increased sun exposure, rather than a direct impact of vitamin D on performance.
5. Dietary Assessment Weaknesses:
The study mentions monitoring dietary intake but provides insufficient detail on how this was done, particularly regarding the reliability of self-reported food diaries. The article notes differences in protein and carbohydrate intake between runners and non-runners; however, there is no discussion of how these dietary factors could independently affect performance outcomes. A more thorough examination of nutrient intake, particularly carbohydrates and proteins, which are essential for endurance athletes, is lacking.
Furthermore, the authors report that vitamin D intake from dietary sources is similar between groups, yet they do not provide a clear hypothesis as to why dietary vitamin D should be expected to correlate with performance in the absence of supplementation.
6. Statistical Power and Sample Size:
The sample size is relatively small (n=45), which raises concerns about the study's ability to detect meaningful differences, particularly when dividing participants into subgroups (e.g., optimal vs. insufficient vitamin D levels). The absence of significant findings for several key variables may be indicative of inadequate statistical power, rather than an absence of effect.
Moreover, the authors place considerable emphasis on p-values without adequately addressing the clinical relevance of their findings. For instance, while some results achieve statistical significance (e.g., differences in white blood cell counts between runners and non-runners), the practical significance of these differences is not investigated.
The article is not of the requisite scientific rigour for publication. The inconsistent methodology employed in the measurement of VO2max, the speculative and unsupported claims made about the impact of vitamin D on performance, and the failure to control for key variables such as sunlight exposure and training type severely compromise the validity of the findings.
Reviewer 2 Report
Comments and Suggestions for Authors
The authors describe a cross-sectional analysis of serum vitamin D levels, blood markers of immune function, diet and physical fitness measures in runners and non-runners.
Overall, the methods should be described in greater detail and the conclusions revised. Some statements need clarification.
Abstract:
Lines 34-35 "The findings suggest that maintaining optimal vitamin D levels is crucial for both athletic performance and immune function." The data presented in the abstract do not support this conclusion. No results on performance are provided and only monocytes (a non-specific marker of immune function) were found to be associated with vitamin D levels. Further, the jump to recommending vitamin D for performance and health is inappropriate based on your findings alone.
Introduction:
Line 54: Check transcription rather than translation.
64: comma needed after poor diet.
106: What is meant by "integration?"
110-115: The first sentence is not consistent with the next statement about vitamin D supplementation having no effect on outcomes.
129: Please clearly state your hypothesis leading to this study.
Methods:
Describe recruitment and provide a flow diagram showing initial responses, screening, declines to participate, exclusions and participants with incomplete data.
How was sample size calculated. What specific outcome was power analysis based upon?
Dietary monitoring: Explain why both food diaries and daily recalls were employed. Who conducted the recalls and were food diaries reviewed with participants? How were discrepancies between the two methods handled? That is, were foods reported in only one method accepted as true? Were reported intakes averaged between the methods?
Explain why outliers, especially for CRP, were not excluded. A CRP of 10 is clearly abnormal.
Figure 2. Interesting findings, but the comparison of vitamin D-deficient vs. sufficient seems off-topic of your main comparison of runners vs. non-runners.
Physical performance is mentioned in the title but received almost no description in the results. It is unclear why this was included in your study.
Discussion:
467-9: The statement about the dietary reference value and recommended "exogenous intake" is unclear. Is not the dietary recommendation meant to be met by dietary intake?
485-6: "The negative relationship between vitamin D levels and monocytes indicates the important role of this vitamin in regulating the immune system and preventing inflammatory status." This is not supported simply by an observed association.
Comments on the Quality of English Language
Errors exist and editing is needed.
Round 2
Reviewer 1 Report
Comments and Suggestions for Authors
Congratulations you have well improved your manuscript
Author Response
|
Response to Academic Editor Comments
|
||
|
1. Summary |
|
|
|
Thank you for the constructive feedback on our manuscript, titled: "A Snapshot of Vitamin D Status, Performance, Blood Markers, and Dietary Habits in Runners and Non-Runners." We appreciate the Academic Editor’s insightful suggestions and their attention to detail in highlighting areas for improvement. This response document addresses the concerns raised, including separated data presentation for males and females and the corresponding analyses and discussion. We hope these revisions and clarifications address the Academic Editor’s concerns and strengthen the scientific rigor of the manuscript. Below, we provide point-by-point responses to each comment. All methodological and results changes have been edited on the new manuscript and shown in red.
|
||
|
2. Point-by-point response to Comments and Suggestions for Authors |
||
|
Comment 1: I have some major concerns regarding the authors' methodology.
Line 144: Please specify the number of males and females in both the runner and non-runner groups. |
||
|
Response 1: We thank the Academic Editor for this comment. This information has been added to the manuscript.
|
||
|
Comment 2: Table 1: Why have the authors combined males and females in this table? Given the known physiological differences, some traits are likely to vary significantly between males and females, such as height, weight, BMI, VO2max, strength, and jumping performance. Were there any differences in 25(OH)D levels between males and females? A combined column (n=45) is unnecessary. Instead, please present the data separately for males and females within each running category and calculate p-values accordingly. |
||
|
Response 2: We thank the Academic Editor for this comment and we appreciate their valuable contributions to improving our study. A two-way multivariate analysis of variance (MANOVA) was conducted to evaluate the differences in the dependent variables according to the group (runners vs. non-runners), gender (male vs. female), and the interaction between gender and group. This analysis confirmed our previous findings and provided the additional information requested. Consequently, the statistical analysis section has been updated to include the MANOVA details. Additionally, Table 1 has been modified to show groups segmented by runners and non-runners, as well as by males and females, along with their respective p-values. As per the editor's recommendation, the combined column has been removed.
|
||
|
Comment 3: Results: If males and females are combined, adjustments for sex and age are required. Otherwise, all analyses (comparisons/relationships) should be conducted separately for males and females. |
||
|
Response 3: We thank the Academic Editor for this comment. We have incorporated these recommendations and adjusted our analyses to account for sex and age.
|
||
|
Comments 4: Table 2: Please separate the data by sex. |
||
|
Response 4: We thank the Academic Editor for this comment. However, we kindly disagree and we prefer to maintain Table 2 without discriminating the gender. In fact, after performing the new analyses using a two-way multivariate analysis, we found no gender interaction with respect to vitamin D values between runners and non-runners. Since vitamin D was the primary outcome of this study, we focused on assessing differences in dietary habits between runners and non-runners. Furthermore, no gender interaction was found for any variables examined, reinforcing the consistency of our primary analysis aimed to analyze differences between runners and non-runners. Finally, as suggested by the academic editor, we removed the combined column in the table 2.
|
||
|
Comments 5: Table 4: Have you combined runners and non-runners when categorizing vitamin D status? If so, this approach is inappropriate. This analysis should be conducted separately for runners and non-runners and also separately for males and females. |
||
|
Response 5: We appreciate the reviewer's comment and the additional help to improve our study. However, after attempting to perform a three-way multivariate analysis of variance (runners vs. non-runners; optimal vs. insufficient/deficient; and sex: male vs. female), we found that most of the information from this analysis is already contained in the analysis presented in Table 1. Additionally, together with these analyses, the Table 3 (Contingency table according to Vitamin D status) provides sufficient information regarding the status of optimal vs. insufficient/deficient for both male and female runners and non-runners. Therefore, we have decided to eliminate Table 4 as it was deemed redundant, as also suggested by reviewer 2 of the previous round.
|
||
|
Comments 6: Section 3.5, Pearson’s Correlation Results: Please perform these analyses separately for males and females. |
||
|
Response 6: We thank the Academic Editor for this comment. We agree with the suggestions and have performed the analyses separately for males and females as requested.
Comments 7: Section 3.6, Multiple Linear Regression Results: Please adjust for age and sex. Response 7: We thank the academic editor for the valuable suggestions. We have adjusted the multiple linear regression analysis to account for age and performed separate analyses for both males and females. Additionally, we have included the goodness-of-fit metrics for both groups Finally, we have replaced Figure 2 with a more representative illustration showing the correlation between Vitamin D levels and monocyte counts for both females and males
|
||
|
Comments 8: Discussion: Please clarify the novelty of this study. |
||
|
Response 8: We thank the Academic Editor for this comment. Our study provides a unique comparison of vitamin D status, immune markers, and performance metrics between runners and sedentary individuals during the autumn season, offering insight into seasonal effects on these parameters. Additionally, we focus on the relationship between vitamin D levels and immune function markers, including monocytes, white blood cells, and neutrophils. This aspect of the study addresses an emerging area of research with implications for athletic performance and overall health. A novel finding of our work is the identification of monocyte count as a significant predictor of vitamin D levels in both male and female participants, suggesting a potential biomarker for understanding vitamin D’s role in immune modulation across different activity levels. Furthermore, by analyzing outdoor activity and sun exposure in runners, we provide a unique perspective on how physical activity may influence endogenous vitamin D synthesis, expanding knowledge on physical activity’s impact on vitamin D status. This information has been added in the discussion section. |
||
Reviewer 2 Report
Comments and Suggestions for Authors
The authors have addressed my concerns.
Author Response

(The authors gave the same response as above.)
